



# A benchmark dataset of diurnal- and seasonal-scale radiation, heat and CO$_2$ fluxes in a typical East Asian monsoon region

Zexia Duan[1#], Zhiqiu Gao[1,3#], Qing Xu[2], Shaohui Zhou[1], Kai Qin[2*], Yuanjian Yang[1*]

[1]Climate and Weather Disasters Collaborative Innovation Center, Key Laboratory for Aerosol-Cloud-Precipitation of China Meteorological Administration, School of Atmospheric Physics, Nanjing University of Information Science and Technology, Nanjing 210044, China
[2]Jiangsu Key Laboratory of Coal-Based Greenhouse Gas Control and Utilization, School of Environment and Spatial Informatics, China University of Mining and Technology, Xuzhou, 221116, China
[3]State Key Laboratory of Atmospheric Boundary Layer Physics and Atmospheric Chemistry, Institute of Atmospheric Physics, Chinese Academy of Sciences, Beijing, 100029, China
* Correspondence to: Yuanjian Yang (yyj1985@nuist.edu.cn) and Kai Qin (qinkai@cumt.edu.cn)
# The authors contributed equally

**Abstract** A benchmark dataset of radiation, heat and CO$_2$ fluxes is crucial to land–atmosphere interaction research. Due to the rapid urbanization and the development of agriculture, land–atmosphere interaction process over the Yangtze River Delta (YRD) of China, which is the typical East Asian monsoon region, is becoming various and complex. To understand the effects of various land cover changes on land–atmosphere interaction in this region, a comprehensive long-term (2011–2019) in situ observation including 30-min meteorology (air temperature, humidity, pressure, wind speed, and wind direction), surface radiative flux, turbulent heat flux, and CO$_2$ flux was conducted at four sites with two typical surface types (i.e., croplands and suburbs) in the YRD. The dataset shows that all four component radiation components, latent heat flux, sensible heat flux, soil heat flux, and CO$_2$ fluxes varied seasonally and diurnally at four sites. Surface energy fluxes exhibited great differences among the four sites. On an annual basis, for two cropland sites, the dominant consumer of net radiation was latent heat flux. At two suburb sites, latent heating dominates from April to November, whereas sensible heating dominates the other months. This dataset will contribute to multiple research fields, including studying land–atmosphere interaction, improving the boundary-layer parameterization schemes, evaluating remote sensing algorithms, and developing climate models in the typical East Asian monsoon region. The dataset is publicly available at https://doi.org/10.5281/zenodo.6552301, last access: 10 May 2022 (Duan et al., 2022).

## 1 Introduction

Land–atmosphere interaction processes, which control the surface–atmosphere exchanges of water, energy and atmospheric carbon dioxide (CO$_2$) across the atmospheric boundary layer, play a key role in ecosystem processes, hydrologic and biogeochemical cycles, and hence in weather and climate (You et al., 2017; Yang et al., 2019). Previous studies show that climate simulations are especially sensitive to seasonal and diurnal variations in a surface energy partitioning of available energy into sensible ($H$) and latent heat ($\lambda E$) fluxes in numerical models (Gao et al., 2004). However, considerable



uncertainties remain in the land–surface parameters of atmospheric numerical models (Sun et al., 2013). It is found that these

parameters' representation is not optimal when validated against the in situ observations (Cuntz et al., 2016). Thus, a comprehensive and accurate in situ eddy covariance (EC) flux measurement is essential to deepen the understanding of the land–atmosphere dynamics (Rao and Reddy, 2019).

The EC technique can derive direct observation of the land–atmosphere carbon, water, and energy fluxes exchanges, and is regarded as the best and the most trustworthy measurement of turbulent fluxes (Baldocchi, 2003). With the

development of the EC tool, there are more than 500 flux towers distributed in various climate zones (Lee et al., 2020; Kang and Cho, 2021). Under the same climate regions, the radiation budget and energy partitioning were mainly modulated by the surface properties (e.g., albedo, roughness length) and subsequently influence air/surface temperature, humidity, and precipitation (Feddema et al., 2005; Jin and Roy, 2005; Li and Wang, 2019). For example, aerodynamically rougher and optically darker oak savanna absorbed more radiation and had higher $H$ and air temperature than the aerodynamically

smoother and optically brighter annual grassland, although both co-exist in semi-arid regions (Baldocchi and Ma, 2013). In the monsoon region, the land cover shifted from vegetation to the bare soil in the Tibetan Plateau and Inner Mongolia not only modified the local thermal and hydrological behavior, but also weakened East Asian summer monsoon circulation and precipitation (Li and Xue, 2010). Although some researchers have explored the roles of land surface processes in the monsoon system (Xue et al., 2004), studies of field observations remain uncertain in the East Asian monsoon region (Bi et

al., 2007), especially in the Yangtze River Delta (YRD) in Eastern China.

The YRD (114°–122° E and 26°–34° N), located in the typical East Asian monsoon region, is one of the most developed regions in China (Yang et al., 2020a). It occupies only 2.2 % area of China, but represents about 16 % of China's total population (Huang et al., 2021) and contributes approximately 18.5 % of China's Gross Domestic Product in 2014 (Hu et al., 2018). Land cover types in the YRD are diverse, dominated by cropland, built-up areas and mountainous areas (mostly

forests and grasslands) across the northern, central, and southern of the YRD, respectively (Figure 1). Recently, the land cover is more complex in this area as it experienced rapid urbanization, industrialization, and agricultural development (Guo et al., 2016). The extensive urbanization increased the $H$ and $CO_2$ emissions, weakened the net surface long-wave radiation, as well as enhanced surface thermal heating to the atmosphere in this region (Chen and Zhang, 2013; Chen et al., 2016a). These alterations modified the urban surface energy budget and the boundary layer structure (Wang et al., 2020; Yang et al.,

2020b), resulting in distinct urban climate effects (e.g., urban heat island, enhanced heat waves, and flooding, Yang and Wang, 2014; Li et al., 2015). Meanwhile, large-scale vegetation degradation and agricultural activities, e.g., the frequent rotation of crop production with several dry–wet cycles in the northern of YRD, modulated the hydrological processes and subsequently affect the regional climate and East Asian monsoon circulation (Chen et al., 2016a). Thus, it is important to clarify the land–atmosphere interaction over the typical surface types in the YRD.

In the past few years, some atmospheric field experiments have been conducted over various land surface types [e.g.,



croplands (Ge et al., 2018; Dai et al., 2019; Duan et al., 2021a) and urban areas (Ao et al., 2016; Wei et al., 2020)] to examine the temporal and spatial characteristics of the energy balance and greenhouse gases in the YRD. Nevertheless, integrated measurements from field stations in the YRD are still not openly shared, or only very limited data during a specified observation period can be accessed. Although China Meteorological Data Service Center (http://data.cma.cn/en, last access: 30 April 2022) has provided some meteorological data in recent years, not all meteorological stations are equipped with EC sensors. Thus, heat and $CO_2$ flux data are often difficult to obtain for a given location and period of interest due to the scarcity of the EC stations (Flerchinger et al., 2009). To improve the knowledge of energy partitioning and $CO_2$ exchange over the typical land surface in the YRD and to find out which surface energy components exhibit the strongest climate signals, a long-term (2011–2019) and continuous integrated observational dataset of land–atmosphere interaction with a high temporal resolution is now released. The underlying observation network is composed of four stations over the two typical surface types in the YRD. At each site, meteorological conditions, four radiation components, EC measurements and soil hydrothermal are examined. This dataset is provided in a XLSX format which can be easily accessed and used by the atmosphere, hydrology, ecology and cryosphere communities, aiming to facilitate the coherence and continuity in scientific understanding of the interactions among the multi-sphere coupled systems in the YRD. These data will be valuable for validating remote-sensing data products, evaluating meteorological and air-quality models (Tsai et al., 2007), and improving boundary-layer parameterization schemes (Bian et al., 2002; Zhang et al., 2020).

In the present work, we provide a long-term (2011–2019) half-hourly-resolution dataset of the integrated land–atmosphere interaction observations over the two typical landscapes (i.e., cropland and suburb) in the YRD and make this dataset available to the public. Section 2 describes the sites, instruments, and data processing methods. Section 3 quantifies the meteorological, solar radiation, heat, and $CO_2$ fluxes at diurnal and seasonal scales at four sites. The availability of the dataset is introduced in Section 4, and a conclusion is shown in Section 5.

## 2 Materials and methods

### 2.1 Sites

The integrated land–atmosphere interaction observation data shown in the present work were collected at four investigation sites including two cropland (i.e., Shouxian and Dongtai station) and two suburb areas (Xuzhou and Dongshan station) of the YRD (Figure 1), which are hereinafter referred to as SX-cropland, DT-cropland, XZ-suburb, and DS-suburb, respectively.

The XZ-suburb site (34.22° N, 117.14° E; 44 m above sea level, Figure 1a) is located in the northwest corner of the Nanhu Campus of China University of Mining and Technology in Tongshan New District, Jiangsu Province, China. A road is approximately 100 m north of the flux tower with a huge traffic volume. To the north of the road is a park, which is dominantly covered by vegetation and lakes. To the south of the observatory are school buildings (mean height 4.5 m), with



high population density, low traffic flow, and high vegetation coverage. Easterly and southeasterly winds prevailed at this site (Figure 2).

The SX-cropland site (32.44° N, 116.79° E; 27 m above sea level, Figure 1b) is situated at Shouxian Agro-Ecosystem Station in Anhui Province, China. The site is characterized by the flat terrain and is covered with silty clay loam soil. The nearest village is over 500 m far away from the observation site. A rice–wheat rotation system was practiced around the EC flux tower. Over this rice–wheat rotation cropland, winter wheat grows from October to June whereas for summer rice it is from June to September every year (Chen et al., 2015). The dominant wind direction ranged from the southeast in spring to the northwest in winter (Figure 2).

The DT-cropland site (32.76° N, 120.47° E; 4 m above sea level, Figure 1d) is about 45 km west of the East China Sea in Jiangsu Province, China. The soil at the site is predominantly clay. The site is homogeneous with a rotation of summer rice and winter wheat cultivated in the field (Duan et al., 2021a; Li et al., 2017). Here, winter wheat grows from December to May while summer rice grows from June to November every year. The dominant wind direction ranged from the southeast in spring to the northwest in winter at this site (Figure 2).

The DS-suburb site (31.08° N, 120.43° E; 13 m above sea level, Figure 1e) is on the southeast shore of Lake Taihu in Jiangsu Province, China (Wang et al., 2014). The observation site is surrounded by water, cropland, and rural houses. The prevailing wind in this area is southeast in the summer and northeast in the winter (Figure 2, Lee et al., 2014).

 At four sites, aerodynamic roughness length [$z_0$, method of Martano (2000)] showed significant seasonal patterns, with the monthly median values of 0.01−0.09 m for SX-cropland, 0.09−0.38 m for DT-cropland, 0.74−1.32 m for XZ-suburb, and

0.44−1.14 m for DS site, respectively (Figure 3).

## 2.2 Instruments

 All sites are equipped with an EC system for long-term, continuous monitoring of the surface radiation, $H$, $\lambda E$ and $CO_2$ fluxes. Table 1 shows the details of instruments at all four sites. The EC system consists of a three-dimensional sonic anemometer (IRGASON, Campbell Scientific Incorporation, USA at XZ-suburb site; CSAT3, Campbell Scientific

Incorporation, USA at three other sites) and a $CO_2/H_2O$ open-path infrared gas analyzer (EC 150, Campbell Scientific Incorporation, USA at SX-cropland site; LI-7500, LI-COR Biosciences, Inc., USA at DT-cropland and DS-suburb sites; IRGASON, Campbell Scientific Incorporation, USA at XZ-suburb site). The EC measurement height was 2.5 m at SX-cropland, 10 m at DT-cropland, 16.5 m at XZ-suburb, and 20 m at DS-suburb site. The four-component net radiometers (CNR-4, Kipp & Zonen B.V., Delft, the Netherlands) were mounted at 1.5 m for SX-cropland, 3 m for DT-cropland, 26.5 m

for XZ-suburb and 1.5 m above ground level (AGL) for DS-suburb sites, respectively. Additionally, the soil heat flux ($G$, using Hukseflux Thermal Sensors HFP01 heat flux plates) was measured at 0.05 m below the ground surface for the SX-cropland site, 0.05, 0.1, 0.2, and 0.4 m for the DT-cropland site, 0.05 m for XZ-suburb site, and 0.05 and 0.10 m for DS-suburb site. Other measurements including air humidity and air temperature (HMP155A; Vaisala, Inc, Helsinki, Finland at SX-cropland and XZ-suburb sites; HMP 45A; Vaisala, Inc, Helsinki, Finland at DT-cropland site, and HMP45C; Vaisala,



Inc, Helsinki, Finland at DS-suburb site) were at a height of 2.5 m at SX-cropland, 10 m at DT-cropland, 16.5 m at XZ-suburb and 20 m at DS-suburb site. Surface air pressure (PTB110, Vaisala, Inc, Helsinki, Finland) was mounted 2.5 m at SX-cropland, 10 m at DT-cropland, 16.5 m at XZ-suburb, and 20 m AGL at DS-suburb site. All instruments were calibrated before installation. More detailed information about the instruments can be seen in Lee et al. (2014), Duan et al. (2021a), and Duan et al. (2021b).

## 135  2.3 Data processing

Each site was visited biweekly to monthly, to maintain instruments as well as download EC data. To gain the high-quality 30-min-resolution EC data, a series of post-processing steps were performed as follows (see Figure 4):

(1)  The raw 10-Hz EC data, including longitudinal ($u$), lateral ($v$), and vertical ($w$) wind velocities, sonic temperature ($T_s$), and water vapor ($H_2O$) density were sampled by a datalogger (model CR3000, Campbell Scientific Inc.) and

then transformed into 30 min binaries with the Campbell Scientific LoggerNet 4.2.1 software.

(2)  LI-COR EddyPro 6.2.1 software was used to calculate and correct 30 min turbulent fluxes of $H$, $\lambda E$, and $CO_2$ fluxes. The data processing in LI-COR EddyPro 6.2.1 software includes: (a) spike removal based on the algorithm of Vickers and Mahrt (1997), i.e., statistical outliers beyond ±3.5 standard deviation in a running window of 10 values were rejected, except more than three values in a row met this criterion (Schmidt et al., 2012), (b) time delay

compensation, (c) double coordinate rotation for the sonic anemometer tilt correction, (d) spectral correction, (e) virtual temperature correction for $H$ (Lee et al., 2004), and (e)Webb–Pearman–Leuning density fluctuations for $\lambda E$ and $CO_2$ fluxes (Webb et al., 1980).

(3)  Quality Control of eddy covariance measurements includes stationarity test, integrated turbulence characteristics test, and footprint analysis. After these tests, the EddyPro quality flags ranged from ''high quality'' (flag 0) to

''suitable for budget analysis'' (flag 1) to ''discard'' (flag 2).

In this paper, Kljun et al. (2015) footprint model was used to examine the spatial representativeness of the EC fluxes at four sites (Figure 5). The average fetch length of the 90% source area was estimated as 225 m, 800 m, 1035 m, and 1558 m for SX-cropland, DT-cropland, XZ-suburb, and DS-suburb flux tower. Based on the results in Figure 5, the land cover fractions were retrieved from the Google Earth image. The compositions in half-hourly EC 90% probable footprints of the

flux tower were separated into five categories: forest, built-up area, cropland, grassland, and water. As shown in Table 2, cropland was the dominant land cover type at SX-cropland and DT-cropland sites, with a fraction of 94% at both sites. From the analysis of the 30 min 90% footprints during the measurement periods, XZ-suburb site included 53 % built-up area, 31 % grassland, 13 % forest, and 3 % water. Whereas for DS-suburb site, the 30 min EC 90 % probable footprint (Kljun et al. 2015) climatology included a half proportion of built-up area and 50 % cropland (Table 2).

Radiative fluxes and meteorological variables were sampled at 1 Hz by the CR3000 datalogger, from which the 30-min means are estimated. Radiative fluxes are limited to physically plausible thresholds, with nocturnal shortwave radiation forced to 0 W m$^{-2}$ (Michel et al., 2008). Meteorological data quality control checks involve reasonable range, internal



consistency, and temporal and spatial consistency based on the methods in Ren et al. (2015). Finally, the meteorological, radiative, heat, and $CO_2$ fluxes data coverage rates are summarized in Table 3, where the percentage values represent the

proportions of the 30 min high-quality measurements.

## 2.4 Methods

$R_n$ is derived from both incoming (↓) and outgoing (↑) shortwave radiation ($K$) and longwave radiation ($L$):

$$R_n = K_\downarrow + L_\downarrow - K_\uparrow - L_\uparrow, \tag{1}$$

$H$ and $\lambda ET$ are estimated by the EC methods (Kaimal and Finnigan, 1994):

$$H = \rho c_p \overline{w'T'}, \tag{2}$$

$$\lambda E = \lambda \frac{M_w/M_a}{\bar{P}} \, \bar{\rho} \, \overline{w'e'}, \tag{3}$$

where $w'$, $T'$, and $e'$ are the turbulent fluctuations from the mean of the vertical wind velocity (m s$^{-1}$), air temperature (K), and water vapor pressure (hPa), respectively, $\rho$ is the air density (kg m$^{-3}$), $c_p$ is the specific heat capacity of air at constant pressure (J kg$^{-1}$ K$^{-1}$), $\lambda$ is the latent heat of vaporization (J kg$^{-1}$), $M_w$ and $M_a$ are the water and air molar mass (g mol$^{-1}$), $P$ is

the air pressure (hPa).

$CO_2$ flux is calculated as follows (Ohtaki and Matsui, 1982):

$$F_c = \overline{w'c'}, \tag{4}$$

Where $F_c$ is $CO_2$ flux (μmol m$^{-2}$ s$^{-1}$), and $c'$ is the fluctuation in the concentration of $CO_2$.

## 3 Results

To advance the knowledge of the land–atmosphere interaction in the YRD, and facilitate the comparison of the similarities and differences between the two typical land surface types, data during the same period (Year 2016) at four sites were selected for analysis in this paper.

### 3.1 Meteorological conditions

        The seasonal (spring, March−May; summer, June−August; autumn, September−November; and winter,

December−February) dynamics of air temperature ($T$) were obvious across all four sites, with an annual mean varied between 16 °C and 17.5 °C (Figure 6a). The average monthly $T$ was relatively high in summer (26−27 °C) but low in winter (4−7 °C) among four sites. The differences in the $T$ between the four sites were minimal. During the observation period, annual mean wind speed ($WS$) was the highest at the DS-suburb site (~3 m s$^{-1}$) as it was measured at the highest observation height (at 20 m AGL, Figure 6b). The annual mean relative humidity was larger at two cropland sites (i.e. 74 % for SX-

cropland and 80 % for DT-cropland sites) than that at two suburb sites (i.e. both 66 % for XZ-suburb and DS-suburb sites,





Figure 6c). The seasonal variations in air pressure ($P$) were opposite to those in air temperature at all four sites (Figures 6a and d). $P$ was high in winter (1023−1025 hPa) but low in summer (1002−1005 hPa) across all four sites (Figure 6d).

## 3.2 Surface radiation budget


At four sites, four surface radiative fluxes [incoming shortwave radiation ($K_\downarrow$), outgoing shortwave radiation ($K_\uparrow$), incoming longwave radiation ($L_\downarrow$), outgoing longwave radiation ($L_\uparrow$)] and surface albedo varied seasonally (Figure 7) and diurnally (Figure 8) with the solar altitude (You et al., 2017). In addition, the seasonal variations in $K_\downarrow$ received at the surface were also greatly affected by weather and cloud conditions (Duan et al., 2021a; Chen et al., 2016b). For instance, the highest daily mean $K_\downarrow$ was 317 W m$^{-2}$ for SX-cropland, 329 W m$^{-2}$ for DT-cropland, 336 W m$^{-2}$ for DS-suburb sites in May 2016, and 332 W m$^{-2}$ for XZ-suburb site in June 2016, respectively (Figure 7a). From May to June, the weather is sunny


with fewer clouds resulting in the higher daily mean $K_\downarrow$ despite the lower solar altitudes. The daily mean peak values of $K_\uparrow$ were 67 W m$^{-2}$ for SX-cropland, 51 W m$^{-2}$ for DT-cropland, 50 W m$^{-2}$ for XZ-suburb sites, and 73 W m$^{-2}$ for DS-suburb sites, respectively (Figure 7b). $K_\uparrow$ varied with both $K_\downarrow$ and surface albedo (Guo et al., 2016). For example, the highest daily mean $K_\uparrow$ at the SX-cropland site that occurred on 29 November 2016 was mainly due to the high snow albedo (Figure 7e). $L_\downarrow$ largely relies on air temperature, cloud properties, and water vapor (Flerchinger et al., 2009). Thus, daily mean $L_\downarrow$ is smallest


in cold winter but largest in warm and wet summer with the peak daily means of 484, 459, 458, and 450 W m$^{-2}$ for SX-cropland, DT-cropland, XZ-suburb, and DS-suburb sites, respectively (Figure 7c). $L_\uparrow$ is closely related to the surface temperature and emissivity (Chen et al., 2016b). Thus, daily mean $L_\uparrow$ was the largest in summer with the maximum values of 521 W m$^{-2}$ for SX-cropland, 501 W m$^{-2}$ for DT-cropland, 501 W m$^{-2}$ for XZ-suburb, and 516 W m$^{-2}$ for DS-suburb sites, respectively (Figure 7d).


The diurnal cycles of the four radiation components for all months are shown in Figure 8. As expected, $K_\downarrow$ exhibits the strongest amplitude of the diurnal cycle among all four radiation components, ranging between 0 and 675 W m$^{-2}$ for SX-cropland, 0−747 W m$^{-2}$ for DT-Cropland, 0−691 W m$^{-2}$ for XZ-suburb, and 0−847 W m$^{-2}$ for DS-suburb sites, respectively (Figure 8a). $K_\uparrow$ keeps the similar diurnal variation trends with $K_\downarrow$. The monthly mean diurnal maxima of $K_\uparrow$ were 104 W m$^{-2}$ for SX-cropland, 101 W m$^{-2}$ for DT-cropland, 86 W m$^{-2}$ for XZ-suburb, and 178 W m$^{-2}$ for DS-suburb sites, respectively


(Figure 8b). $L_\downarrow$ and $L_\uparrow$ has a smaller diurnal cycle amplitude, with slightly higher values in the afternoon [around 14:00−15:00 Local Time (LT)]. Surface albedo ($K_\uparrow / K_\downarrow$) directly modulates the energy absorbed by the surface, which is mainly influenced by surface conditions, solar angle, and weather conditions (Zhang et al., 2014). The midday (10:00–15:00 LT) albedo varied diurnally between 0.1 and 0.26. The annual mean albedo was 0.163, 0.133, 0.143, and 0.195 for SX-cropland, DT-Cropland, XZ-suburb, and DS-suburb sites, respectively (Figure 8e).



## 3.3 Surface energy fluxes

The surface energy balance fluxes play a key role in regulating the ground thermal regime (Hoelzle et al., 2022). Figure 9 shows the remarkable seasonal variations in daily mean $R_n$, $\lambda E$, $H$, and $G$ at a depth of 0.05 m ($G_{0.05}$). Both $R_n$ and $G_{0.05}$ were high in spring and summer but low in autumn and winter at four sites in the YRD (Figures 9a and 9d). Although in the same climate monsoon area, there exist large differences in $\lambda E$ and $H$ over different surface types. Seasonal variations in $\lambda E$ at two cropland sites have doublet peaks, which were closely related to the crop phenology and agricultural activities (Duan et al., 2021a). For example, daily mean $\lambda E$ firstly increased from 3 W m$^{-2}$ (5 W m$^{-2}$) in January to the peak value of 110 W m$^{-2}$ (182 W m$^{-2}$) in April and then gradually decreased when wheat harvest at the SX-cropland site (DT-cropland site). In mid-June, the rice seedlings were transplanted and daily mean $\lambda E$ attained the second peak of 155 W m$^{-2}$ (224 W m$^{-2}$) in August at the SX-cropland site (DT-cropland site). The extensively irrigated cropland enhances the available energy to support evaporation and results to lower $H$ (Dou et al., 2019). Thus, at two cropland sites, the daily mean $H$ was almost lower than 35 W m$^{-2}$. However, $\lambda E$ have a unimodal distribution at the DS-suburb site in 2016, with the daily mean peaks of 165 W m$^{-2}$ in July (Figure 9b). The daily mean $H$ at XZ-suburb and DS-suburb varied from –21 to 70 W m$^{-2}$ and –39 to 81 W m$^{-2}$, respectively.

Figure 10 shows the significant diurnal dynamics of $R_n$, $\lambda E$, $H$, and $G_{0.05}$ in all months. As expected, $R_n$ peaked at ~13:00 LT during the daytime due to the strong heating of the surface by the sun, while it is negative ($L_\uparrow > L_\downarrow$, with $K_\downarrow = K_\downarrow = 0$, Figure 8) at night when surface radiative cooling dominates (Nelli et al., 2020). The diurnal dynamics of $R_n$ vary from –38 to 528, –38 to 590, –61 to 513, and –64 to 562 W m$^{-2}$ for SX-cropland, DT-cropland, XZ-suburb, and DS-suburb sites, respectively. At two cropland sites, $\lambda E$ is the largest consumer of the $R_n$ in the whole year with the annual mean midday $\lambda E/R_n$ of 35 % and 58 % at SX-cropland and DT-cropland sites. At two suburb sites, the midday $H/R_n$ ranged between 4 % and 55 % (16 % and 70 %) for the XZ-suburb site (the DS-suburb site), while $\lambda E/R_n$ ranged between 4 % and 49 % (29 % and 93 %) for the XZ-suburb site (DS-suburb site), and the largest consumer of $R_n$ shifted between $\lambda E$ and $H$. These seasonal fluctuations in $\lambda E$ and $H$ at two suburb areas were mainly due to the seasonal cycles of the vegetation cover and the intensive human activities (Duan et al., 2021a). The $G_{0.05}$ was small and varied between –20 and 40 W m$^{-2}$ diurnally. Generally, radiation and heat fluxes showed distinct differences under the typical surface types in the monsoon area of the YRD of China.

## 3.4 Carbon flux

Figure 11a exhibits the seasonal variations of the daily mean $CO_2$ flux. At two rice-wheat rotation cropland sites, the $CO_2$ flux measures the photosynthesis and respiration of the crops. At the beginning of the rice growing period (June) the $CO_2$ emission is high at two cropland sites, with the peak daily mean $CO_2$ flux value of 2.4 μmol m$^{-2}$ s$^{-1}$ for SX-cropland and 5.5 μmol m$^{-2}$ s$^{-1}$ for DT-cropland site, respectively. This is mainly caused by $CO_2$ released from aquatic weeds and algae



from the paddy water surface (Nishimura et al., 2015). The daily mean $CO_2$ flux reaches its minima in August (–11 μmol m$^{-2}$ s$^{-1}$ for SX-cropland and –10 μmol m$^{-2}$ s$^{-1}$ for DT-cropland site) when the rice photosynthetic rates are quite strong. After this, the rice leaves gradually turn yellow and tend to mature. The wheat field had similar patterns to the rice paddy. At the XZ-suburb site, the daily mean $CO_2$ flux was almost positive in spring, autumn and winter, and the maximum daily mean value can reach 6.5 μmol m$^{-2}$ s$^{-1}$, indicating that human activities (e.g., greater residential heating) dominant this period. At the DS-suburb site, daily mean $CO_2$ flux varied between –4 and 3 μmol m$^{-2}$ s$^{-1}$, with high values in May and low values in September.

A marked and significant diurnal cycle of $CO_2$ flux is shown in Figure 11b. Positive nocturnal values, respectively reaching at 6.4, 8.2, 8.2, 4.5 μmol m$^{-2}$ s$^{-1}$ for SX-cropland, DT-Cropland, XZ-suburb, and DS-suburb sites, which were mainly related to the poor night-time atmospheric mixing (Cheng et al., 2018), lower boundary layer height (Hassan, 2015), plant respiration (Mai et al., 2020), and anthropogenic sources (Hu et al., 2018). Mid-afternoon negative $CO_2$ fluxes (about –8–29 μmol m$^{-2}$ s$^{-1}$, negative values refer that the ecosystem absorbs $CO_2$ from the atmosphere) at SX-cropland, DT-Cropland, XZ-suburb, and DS-suburb sites were due to the active biospheric photosynthesis and favorable dispersion conditions (Grimmond et al., 2002). However, the $CO_2$ flux almost remain positive in January and December; i.e. suburb surface is a net $CO_2$ source. Vegetation in XZ-suburb has a clear effect during the daytime, but it is not enough to offset the strong anthropogenic emissions, which are significant during rush hours in the morning and afternoon in Figure 11b.

## 4 Data availability

All datasets presented in this paper are freely available at https://doi.org/10.5281/zenodo.6552301, last access: 10 May 2022 (Duan et al., 2022).

## 5 Conclusion

The turbulent flux parameters in current numerical models suffer from poor representation in the monsoon region, especially in the YRD, as it is experiencing rapid land-use changes. Thus, continuous (2011–2019) and high-quality land–atmosphere interaction observations are collected for a deeper understanding of the land surface processes in the YRD. In this paper, field measurements over two typical underlying surfaces, i.e. cropland and suburb surface at four sites in the monsoonal YRD region were presented. Our findings show that the individual radiation components, $H$, $\lambda E$, $G$, and $CO_2$ fluxes exhibit diurnal and seasonal variations, which also depended on the local underlying surface conditions. Over the year, $\lambda E$ dominates the land–atmosphere heat flux exchange at two SX-cropland and DT-cropland sites. At two suburb sites, however, the dominant consumer of the $R_n$ fluctuated between $\lambda E$ and $H$, which could subsequently modulate the local climate.



Generally, this dataset provides comprehensive, high temporal resolution and high-quality in situ flux observations in the YRD, which is valuable for studying land–atmosphere interactions and their impacts on weather change research. In addition, this dataset could provide accurate parameters and calibrations for reanalysis data, remote sensing products, and climate models.

**Author contributions**

GZ, QK, and YY designed the experiments and carried them out. DZ, XQ, and ZS performed data processing, organization, and figure generation. DZ and YY wrote the manuscript, and all authors participated in the revision of the paper.

**Competing interests**

The authors declare that they have no conflict of interest.

**Acknowledgments**

We sincerely thank all the scientists, engineers, and students who participated in the field campaigns, maintained the measurement instruments, and processed the observations.

**Financial support**

This work was funded by the National Natural Science Foundation of China (Grant: 41875013), and the Postgraduate
Research & Practice Innovation Program of Jiangsu Province (KYCX21_0950).

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



**Table 1. Descriptions of the sensors used at four stations.**

| Instrument | Site name | Variable (unit) | Model and manufacturer | Installation hight (m) |
|---|---|---|---|---|
| Three-dimensional sonic anemometer | SX-cropland | three-dimensional wind speed (m s$^{-1}$), sonic air temperature (℃) | CSAT3, Campbell | 2.5 |
| | DT-cropland | | CSAT3, Campbell | 10 |
| | XZ-suburb | | IRGASON, Campbell | 16.5 |
| | DS-suburb | | CSAT3, Campbell | 20 |
| $CO_2$/$H_2O$ open-path infrared gas analyzer | SX-cropland | $CO_2$ (mg m$^{-3}$), $H_2O$ (mg m$^{-3}$) | EC150, Campbell | 2.5 |
| | DT-cropland | | LI-7500, LI-COR | 10 |
| | XZ-suburb | | IRGASON, Campbell | 16.5 |
| | DS-suburb | | LI-7500A, LI-COR | 20 |
| Four-component net radiometer | SX-cropland | Radiations (W m$^{-2}$) | CNR4, Kipp & Zonen | 1.5 |
| | DT-cropland | | CNR4, Kipp & Zonen | 3 |
| | XZ-suburb | | CNR4, Kipp & Zonen | 26.5 |
| | DS-suburb | | CNR4, Kipp & Zonen | 1.5 |
| Soil heat flux plate | SX-cropland | Soil heat flux (W m$^{-2}$) | HFP01, Hukseflux | −0.05 |
| | DT-cropland | | HFP01, Hukseflux | −0.05, −0.1, −0.2, and −0.4 |
| | XZ-suburb | | HFP01, Hukseflux | −0.05 |
| | DS-suburb | | HFP01, Hukseflux | −0.05 and −0.10 |
| Surface atmospheric pressure sensor | SX-cropland | Pressure (hPa) | PTB110, Vaisala | 2.5 |
| | DT-cropland | | PTB110, Vaisala | 10 |
| | XZ-suburb | | PTB110, Vaisala | 16.5 |
| | DS-suburb | | PTB110, Vaisala | 20 |
| Air temperature and humidity | SX-cropland | Air temperature (℃) and humidity (%) | HMP155A, Vaisala | 2.5 |
| | DT-cropland | | HMP45A, Vaisala | 10 |
| | XZ-suburb | | HMP155A, Vaisala | 16.5 |
| | DS-suburb | | HMP45C, Vaisala | 20 |

**Table 2. Land cover fractions within the 90 % footprints at four sites.**

| Site ID | | SX-cropland | DT-cropland | XZ-suburb | DS-suburb |
|---|---|---|---|---|---|
| Land cover fraction within the 90 % source area | Forest | 0.00 | 0.00 | 0.13 | 0.00 |
| | Built-up area | 0.04 | 0.06 | 0.53 | 0.50 |
| | Cropland | 0.94 | 0.94 | 0.00 | 0.50 |
| | Grassland | 0.00 | 0.00 | 0.31 | 0.00 |
| | Water | 0.02 | 0.00 | 0.03 | 0.00 |



**Table 3. The proportion of data availability. The percentage represents the proportion of 30 min high-quality EC**
**data.**

| Variable type | SX-cropland | | DT-cropland | | XZ-suburb | | DS-suburb | |
|---|---|---|---|---|---|---|---|---|
| | Duration | Proportion | Duration | Proportion | Duration | Proportion | Duration | Proportion |
| Wind direction | | 0.99 | | 0.99 | | 0.82 | | 0.92 |
| Wind speed | | 0.99 | | 0.99 | | 0.82 | | 0.92 |
| Air temperature | | 0.99 | | 0.99 | | 0.87 | | 0.92 |
| Relative humidity | | 0.96 | | 0.99 | | 0.87 | | 0.92 |
| Air pressure | | 0.99 | | 0.99 | | 0.87 | | 0.92 |
| $K_\downarrow$ | | 0.90 | | 0.96 | | 0.67 | | 0.93 |
| $K_\uparrow$ | 15 Jul 2015–24 Apr 2019 | 0.90 | 1 Dec 2014–30 Nov 2017 | 0.96 | 27 Mar 2014–22 Jan 2017 | 0.67 | 16 Apr 2011–1 Jan 2019 | 0.87 |
| $L_\downarrow$ | | 0.90 | | 0.96 | | 0.65 | | 0.75 |
| $L_\uparrow$ | | 0.90 | | 0.96 | | 0.67 | | 0.76 |
| $\lambda E$ | | 0.71 | | 0.70 | | 0.72 | | 0.76 |
| $H$ | | 0.87 | | 0.78 | | 0.72 | | 0.80 |
| $G_{0.05}$ | | 0.90 | | 0.96 | | 0.87 | | 0.94 |
| $CO_2$ | | 0.73 | | 0.71 | | 0.70 | | 0.82 |



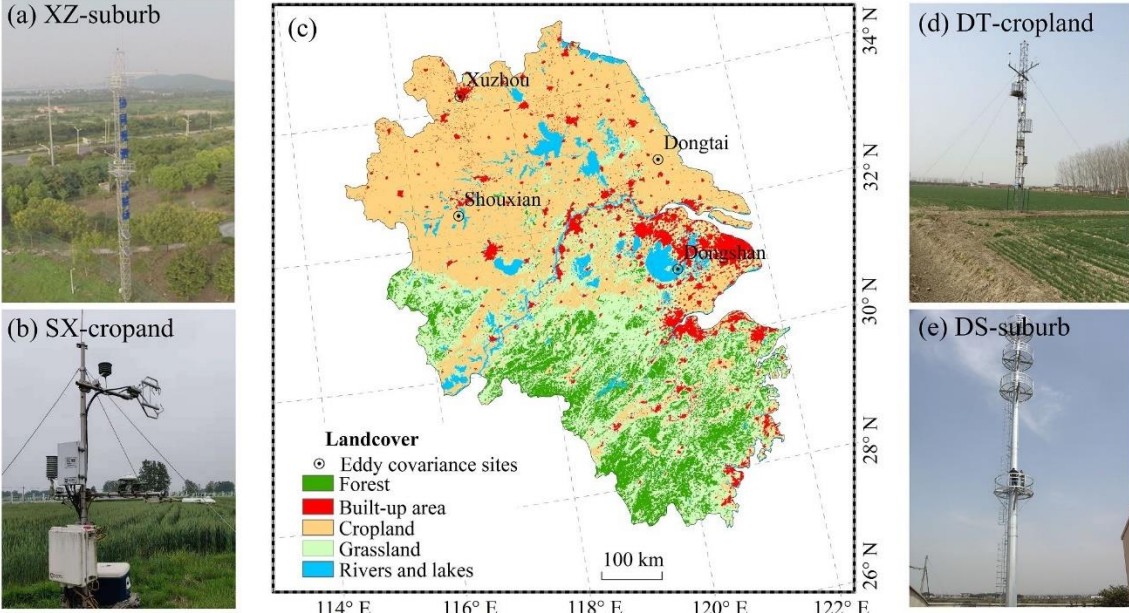

**Figure 1: Surface types of the four field sites at (a) XZ-cropland, (b) SX-cropland, (d) DT-suburb, (e) DS-suburb, and (c) the relative position in the Yangtze River Delta of Eastern China.**





**Figure 2: The seasonal (spring, March−May; summer, June−August; autumn, September−November; and winter, December−February) dynamics of wind roses (22.5° bins, 30 min data) stratified by wind speed frequency for (a)−(d) SX-cropland, (e)−(h) DT-cropland, (i)−(l) XZ-suburb, and (m)−(p) DS-suburb sites.**




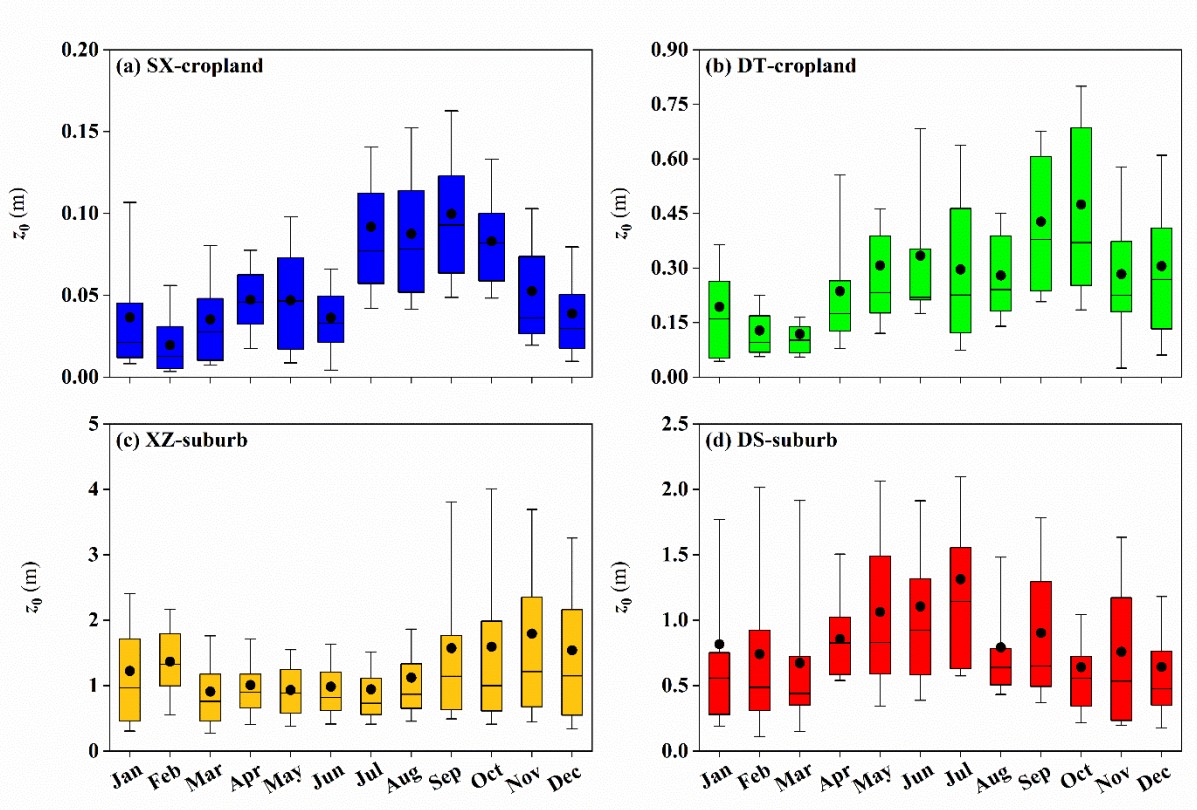

**Figure 3: Variations of monthly aerodynamic roughness length ($z_0$) at (a) SX-cropland, (b) DT-cropland, (c) XZ-suburb, and (d) DS-suburb sites. Boxplots (25, 50 and 75th percentiles) with 10 and 90th percentiles whiskers plus mean (black dot).**






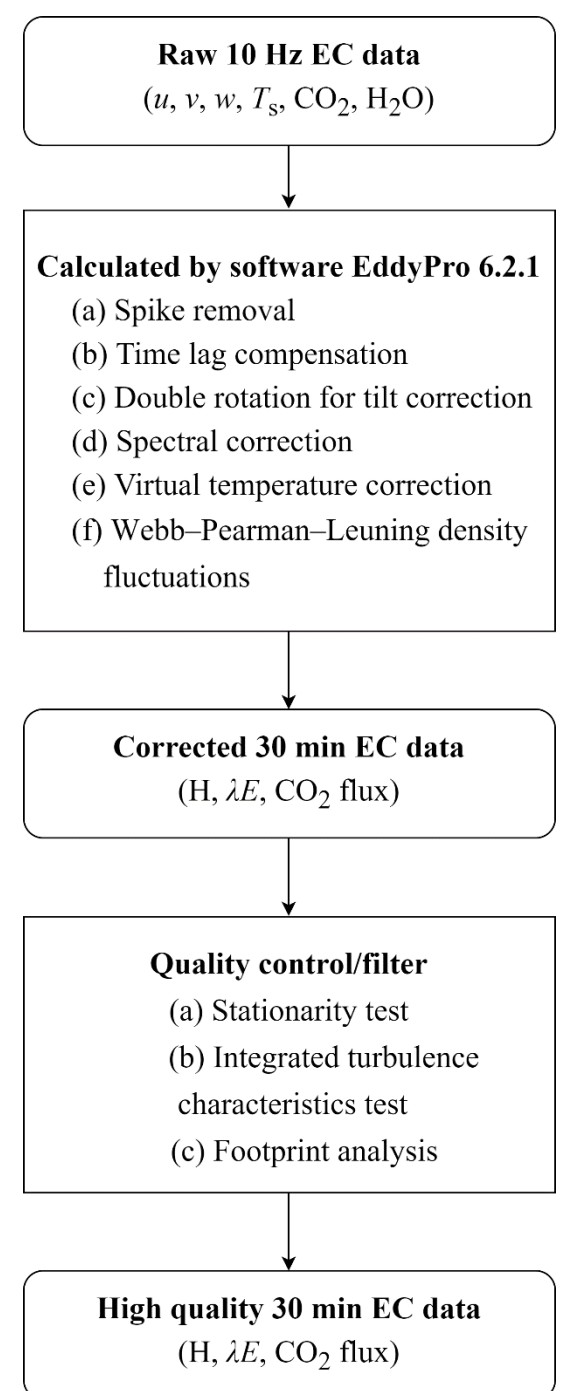

**Figure 4: Flowchart of the EC data processing and quality assurance and control.**

**Figure 5: Probable eddy covariance flux source areas (70 % and 90 %, yellow lines from inside to outside) by Kljun et al. (2015) footprint model for (a) SX-cropland, (b) DT-cropland, (c) XZ-suburb, and (d) DS-suburb sites. The red star represents the flux tower location.**






**Figure 6: Monthly mean (a) air temperature (*T*), (b) wind speed (*WS*), (c) relative humidity (*RH*), and (d) air pressure (*P*).**





**Figure 7: Seasonal variations in daily mean (a) incoming shortwave radiation $K_\downarrow$, (b) outgoing shortwave radiation $K_\uparrow$, (c) incoming longwave radiation $L_\downarrow$, (d) outgoing longwave radiation $L_\uparrow$, and (e) surface albedo.**

Earth System
**Science**
**Data**

Open Access    Discussions

**Figure 8: Diurnal cycle of the monthly mean (a) incoming shortwave radiation $K_\downarrow$, (b) outgoing shortwave radiation $K_\uparrow$, (c) incoming longwave radiation $L_\downarrow$, (d) outgoing longwave radiation $L_\uparrow$, and (e) surface albedo.**

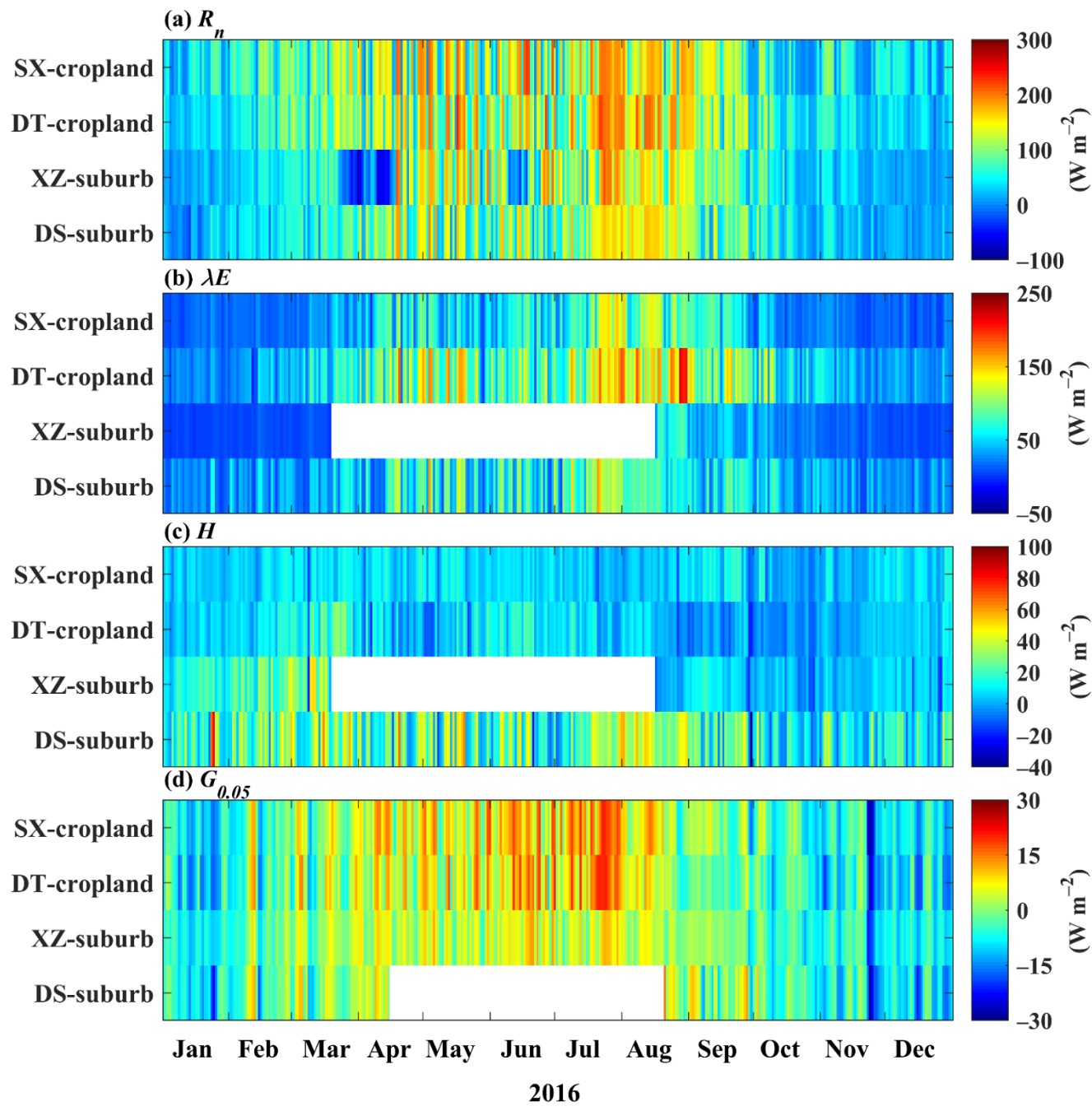

**Figure 9: As in Figure 7, but for (a) $R_n$, (b) $\lambda E$, (c) $H$, and (d) $G_{0.05}$.**



**540** **Figure 10: Diurnal cycle of the monthly mean net radiation ($R_n$), latent heat flux ($\lambda E$), sensible heat flux ($H$), and soil heat flux at a depth of 0.05 m ($G_{0.05}$) at (a) SX-cropland, (b) DT-cropland, (c) XZ-suburb, and (d) DS-suburb sites.**



Figure 11: (a) Seasonal and (b) monthly diurnal patterns of CO₂ exchanges in 2016 at four sites.