# Peer review of "A benchmark dataset of diurnal- and seasonal-scale radiation, heat and CO2 fluxes in a typical East Asian monsoon region"

_Earth System Science Data, 2022_

## Author Comment (AC1)

**Response to Reviewer 1**

Dear Reviewer,

We greatly appreciate your efforts and your helpful comments in reviewing our article. We have incorporated all of your comments in the revised manuscript.

We respond below in blue to your comments item-by-item.

Sincerely yours,

Yuanjian Yang and Kai Qin on behalf of all co-authors

**Reviewer #1: General comment**

The manuscript by Duan et al describes a significant database of eddy-covariance and micro-meteorological measurements in a typical East Asian monsoon region of China. The data quality control for EC data is introduced in detail, and the manuscript also presents the variations of each variable at diurnal, daily and monthly scales, to some extent, indicating that the data accuracy is reasonable. This dataset will contribute to multiple research fields, including studying land‐atmosphere interaction, improving the boundary-layer parameterization schemes, evaluating remote sensing algorithms, and developing climate models in the typical East Asian monsoon region. The manuscript is well written and documented, and I suggest it will be published after some revision.

*Response:* Many thanks for your positive comments.

**Major comments:**

[1] The description of the time used is unclear in the data file. Is it local time or UTC?

*Response:* Thank you very much for pointing this out. We have added this description in the "Data availability" section as follows (Line 277): "The local time (UTC+8) was used at four sites."

[2] For long term data sets, sensor calibration is important, especially for radiation measurements. The sensor calibration in the current Manuscript may need to be supplemented.

*Response:* Thank you very much for your kind suggestion. More information about sensor calibration has been supplemented as follows (Lines 132–136):

"All instruments were calibrated by professional engineers. For example, the calibration steps of $CO_2/H_2O$ open-path infrared gas analyzer mainly included: (a) determining the calibration coefficients and (b) setting zero and span. The three-dimensional sonic anemometer needed a factory calibration (i.e., test for wind offset and check for diagnostic flags) every two years, while the four-component net radiometer was calibrated every year to guarantee the radiation data quality."

[3] Define the radiation, turbulent, and $CO_2$ flux direction in section 2.4.

*Response:* Thank you very much for pointing this out. We have defined the radiation, turbulent, and $CO_2$ flux direction in section 2.4 as follows:

Line 172: "$R_n$ (positive radiation toward the surface) is derived from incoming ($\downarrow$, downward) and outgoing ($\uparrow$, upward) components of shortwave radiation ($K$) and longwave radiation ($L$):"

Line 175: "$H$ and $\lambda E$ (positive flux away from the surface) are estimated by the EC methods (Kaimal and Finnigan, 1994):"

Line 182: "$F_c$ (positive $CO_2$ flux away from the surface) is calculated as follows (Ohtaki and Matsui, 1982):"

**Minor comments:**

Line 20: "four component radiation components" should be "four radiation components".

*Response:* Corrected. (Line 20)

Line 43: replace the word "influence" with "influenced".

*Response:* Corrected. (Line 42)

Line 45: change the word "both co-exist" with "both of them co-existed".

*Response:* Corrected. (Line 45)

Lines 69–72: the sentence is better to modified as follows: "Although China Meteorological Data Service Center (http://data.cma.cn/en, last access: 30 April 2022) has provided some meteorological data in recent years, EC sensors have not been commonly equipped in those meteorological stations, making it difficult to obtain heat and $CO_2$ flux data at some specific places or periods (Flerchinger et al., 2009)."

*Response:* Corrected. (Lines 70–73)

Line 151: change "Kljun et al. (2015) footprint model" to "the footprint model proposed by Kljun et al. (2015)".

*Response:* Corrected. (Line 156)

Line 174: add the word "and" before "$P$ is the air pressure (hPa)".

*Response:* Corrected. (Line 180)

Line 178: "Where" should be "where".

*Response:* Corrected. (Line 184)

Figure 7e: The unit of albedo should not be "W m$^{-2}$".

*Response:* Figure 7e has been revised as follows:

[Figure]

---

## Author Comment (AC2)

**Reviewer #2:** **General comment**

The authors produce a long-term eddy-covariance data set from two wheat-rice rotation cropland sites and two suburb sites in a typical East Asian monsoon region of Eastern China. They present and evaluate the general meteorological data, radiation data, turbulent fluxes, and $CO_2$ fluxes. The descriptions of the sites and methods are clear. They provide a valuable dataset, and the results are publishable. Therefore, I would like to suggest some adjustments that may help improve the study.

*Response:* Many thanks for your positive and valuable comments, and they were very helpful in refining the manuscript. According to your suggestions, we have supplied additional analyses to further substantiate our findings. We hope that this revision could address all your concerns in a satisfying manner. Now, we are responding these comments (in blue) item-by-item.

**Major comments:** As a descriptive manuscript related to the field monitoring data, it should present the details about the instruments and data process as much as possible. For example, what are the operating range, accuracy, and precision of the sensors used? How to process the data gaps in the datasets? Was in-filling performed on these gaps?

*Response:* Many thanks for your constructive comments. In the revised manuscript, we have provided more details.

a.  The operating range and accuracy of the sensors used at four sites have been added in Table 1.

**Table 1. Descriptions of the sensors used at four stations.**

| Instrument | Site name | Variable (unit) | Model and manufacturer | Installation hight (m) | Measurement range | Accuracy |
|---|---|---|---|---|---|---|
| Three-dimensional sonic anemometer | SX-cropland | three-dimensional wind speed ($u_x$, $u_y$ and $u_z$ m s$^{-1}$), sonic air temperature ($T_s$, °C) | CSAT3, Campbell | 2.5 | $u$ and $v$: −65 to 65 m s$^{-1}$ $w$: −65 to 65 m s$^{-1}$ $T_s$: −50 °C to 60 °C | $u$ and $v$: ±0.04 m s$^{-1}$ $w$: ±0.02 mm s$^{-1}$ $T_s$: ±0.025 °C |
| | DT-cropland | | CSAT3, Campbell | 10 | | |
| | XZ-suburb | | IRGASON, Campbell | 16.5 | $u$ and $v$: −65 to 65 m s$^{-1}$ $w$: −65 to 65 m s$^{-1}$ $T_s$: −50 °C to 60 °C | $u$ and $v$: ±0.08 m s$^{-1}$ $w$: ±0.04 mm s$^{-1}$ $T_s$: ±0.025 °C |
| | DS-suburb | | CSAT3, Campbell | 20 | $u$ and $v$: −65 to 65 m s$^{-1}$ $w$: −65 to 65 m s$^{-1}$ $T_s$: −50 °C to 60 °C | $u$ and $v$: ±0.04 m s$^{-1}$ $w$: ±0.02 mm s$^{-1}$ $T_s$: ±0.025 °C |
| $CO_2$/$H_2O$ open-path infrared gas analyzer | SX-cropland | $CO_2$ (mg m$^{-3}$), $H_2O$ (mg m$^{-3}$) | EC150, Campbell | 2.5 | $CO_2$: 0 to 1830 mg m$^{-3}$ $H_2O$: 0 to 44 g m$^{-3}$ | $CO_2$: < 1% $H_2O$: <2% |
| | DT-cropland | | LI-7500, LI-COR | 10 | $CO_2$: 0 to 5148 mg m$^{-3}$ $H_2O$: 0 to 42 g m$^{-3}$ | $CO_2$: ±0.2 mg m$^{-3}$ $H_2O$: ±0.02 g m$^{-3}$ |
| | XZ-suburb | | IRGASON, Campbell | 16.5 | $CO_2$: 0 to 1830 mg m$^{-3}$ $H_2O$: 0 to 44 g m$^{-3}$ | $CO_2$: 0.2 mg m$^{-3}$ $H_2O$: 0.0035 g m$^{-3}$ |
| | DS-suburb | | LI-7500A, LI-COR | 20 | $CO_2$: 0 to 5148 mg m$^{-3}$ $H_2O$: 0 to 42 g m$^{-3}$ | $CO_2$: ±0.2 mg m$^{-3}$ $H_2O$: ±0.02 g m$^{-3}$ |
| Four-component net radiometer | SX-cropland | Radiations (W m$^{-2}$) | CNR4, Kipp & Zonen | 1.5 | $K_\downarrow$ and $K_\uparrow$: 0.3 to 2.8 μm $L_\downarrow$ and $L_\uparrow$: 4.5 to 42 μm | $K_\downarrow$ and $K_\uparrow$: 5 to 20 μV W$^{-1}$ m$^{-2}$ $L_\downarrow$ and $L_\uparrow$: 5 to 15 μV W$^{-1}$ m$^{-2}$ |
| | DT-cropland | | CNR4, Kipp & Zonen | 3 | | |
| | XZ-suburb | | CNR4, Kipp & Zonen | 26.5 | | |
| | DS-suburb | | CNR4, Kipp & | 1.5 | | |

| | | | | Zonen | | |
|---|---|---|---|---|---|---|
| Soil heat flux plate | SX-cropland | Soil heat flux (W m$^{-2}$) | HFP01, Hukseflux | –0.05 | $\pm$2000 W m$^{-2}$ | $\pm$3% |
| | DT-cropland | | HFP01, Hukseflux | –0.05, –0.1, –0.2, and –0.4 | | |
| | XZ-suburb | | HFP01, Hukseflux | –0.05 | | |
| | DS-suburb | | HFP01, Hukseflux | –0.05 and –0.10 | | |
| Surface atmospheric pressure sensor | SX-cropland | Pressure ($P$, hPa) | PTB110, Vaisala | 2.5 | $P$: 500 to 1100 hPa | $\pm$0.3 hPa (20 °C) |
| | DT-cropland | | PTB110, Vaisala | 10 | | |
| | XZ-suburb | | PTB110, Vaisala | 16.5 | | |
| | DS-suburb | | PTB110, Vaisala | 20 | | |
| Air temperature and humidity | SX-cropland | Air temperature ($T_a$, °C) and humidity (RH, %) | HMP155A, Vaisala | 2.5 | $T_a$: –80 to +60 °C RH: 0.8 to 100 % | Depends on $T_a$ and $RH$ |
| | DT-cropland | | HMP45A, Vaisala | 10 | $T_a$: –40 to +60 °C RH: 0 to 100 % | $T_a$: $\pm$0.2 °C (20 °C) $RH$: $\pm$2% (0–90%) $\pm$3% (90–100%) |
| | XZ-suburb | | HMP155A, Vaisala | 16.5 | $T_a$: –80 to +60 °C RH: 0.8 to 100 % | Depends on $T_a$ and $RH$ |
| | DS-suburb | | HMP45C, Vaisala | 20 | $T_a$: –40 to +60 °C RH: 0 to 100 % | $T_a$: $\pm$0.2 °C (20 °C) $RH$: $\pm$2% (0–90%) $\pm$3% (90–100%) |

b. In terms of data gaps and gap-filling process in the dataset, detailed descriptions were added in Section 2.3 as follows (Lines 153–155):

"Note that EC data under unfavorable weather conditions (e.g., rainy and foggy days) or during periods of instrument malfunctions were excluded. However, there was no interpolation in the measurement, which can keep initial information of observations."

**Minor comments:**

Lines 123–125: The installation height of the four-component net radiometers at SX-cropland and DS-suburb sites were same, please write them together as follows: "The four-component net radiometers (CNR-4, Kipp & Zonen B.V., Delft, the Netherlands) were mounted at 1.5 m for SX-cropland and DS-suburb, 3 m for DT-cropland, and 26.5 m above ground level (AGL) for XZ-suburb sites, respectively."

*Response:* Corrected. (Line 124–126)

Lines 128–132: Similarly, the installation heights at four sites for air humidity, air temperature and surface air pressure were the same, please combine them.

*Response:* Many thanks for your kind comments. We have revised this sentence as follows (Line 129–132): "Other measurements including air humidity and air temperature (HMP155A; Vaisala, Inc, Helsinki, Finland at SX-cropland and XZ-suburb sites; HMP 45A; Vaisala, Inc, Helsinki, Finland at DT-cropland site, and HMP45C; Vaisala, Inc, Helsinki, Finland at DS-suburb site) and surface air pressure (PTB110, Vaisala, Inc, Helsinki, Finland) were at a height of 2.5 m at SX-cropland, 10 m at DT-cropland, 16.5 m at XZ-suburb and 20 m at DS-suburb site."

Line 169: change "$\lambda ET$" to "$\lambda E$".

*Response:* Corrected. (Line 175)

Lines 277–280: the sentence is better to modified as follows: "Take the Year 2016 as an example, $\lambda E$ dominates the land–atmosphere heat flux exchange at two cropland sites (SX-cropland and DT-cropland). However, the dominant consumer of the $R_n$ fluctuated between $\lambda E$ and $H$ at two suburb sites (XZ-suburb and DS-suburb), which could subsequently modulate the local climate."

*Response:* We have revised this sentence as follows (Lines 285–287): "Take the Year of 2016 as an example, $\lambda E$ dominated the land–atmosphere heat flux exchange at two cropland sites (SX-cropland and DT-cropland). However, the dominant consumer of the $R_n$ fluctuated between $\lambda E$ and $H$ at two suburb sites (XZ-suburb and DS-suburb), which could subsequently modulate the local climate."

Lines 260 and 263: "DT-Cropland" should be "DT-cropland".

*Response:* Corrected. (Line 267 and 270)

Figure 1b: change the label "SX-cropand" to "SX-cropland".

*Response:* Figure 1 has been revised as follows:

[Figure]